# Development of Novel Detection Method for Rutinosidase in Tartary Buckwheat (*Fagopyrum tataricum* Gaertn.)

**DOI:** 10.3390/plants11030320

**Published:** 2022-01-25

**Authors:** Tatsuro Suzuki, Toshikazu Morishita, Shigenobu Takigawa, Takahiro Noda, Koji Ishiguro, Shiori Otsuka

**Affiliations:** 1Kyushu Okinawa Agricultural Research Center, National Agriculture and Food Research Organization, Suya 2421, Koshi, Kumamoto 861-1192, Japan; 2Radiation Breeding Division, Institute of Crop Science, National Agriculture and Food Research Organization, 2425 Kamimurata, Hitachiomiya, Ibaraki 319-2293, Japan; tosikazu@affrc.go.jp; 3Hokkaido Agricultural Research Center, National Agriculture and Food Research Organization, Hitsujigaoka, Toyohira, Hokkaido, Sapporo 062-8555, Japan; takigawa@affrc.go.jp; 4Hokkaido Agricultural Research Center, National Agriculture and Food Research Organization, Shinsei, Memuro, Hokkaido, Kasai-Gun 082-0081, Japan; noda@affrc.go.jp (T.N.); kuro@affrc.go.jp (K.I.); otsukas517@affrc.go.jp (S.O.)

**Keywords:** tartary buckwheat, rutin, rutinosidase, detection, bitterness, alum

## Abstract

Contamination of rutinosidase is a major problem for rutin-rich food due to the hydrolysis of the functional compound rutin and generation of strong bitterness caused by the hydrolyzed moiety quercetin. This problem sometimes occurs between the trace and normal rutinosidase Tartary buckwheat varieties. Here, we developed a simple and rapid method for rutinosidase detection in ‘ripening seeds using UV light’ and in ‘dough using alum-flavonoid complexation’ from Tartary buckwheat (*Fagopyrum tataricum* Gaertn.). Normal rutinosidase seeds can be distinguished from trace-rutinosidase mature seeds and ripening young seeds using a rutin solution by comparing the muddiness, which corresponds to quercetin generation. In dough, we detected a threefold relative increase in rutinosidase activity corresponding to 1% contamination of normal rutinosidase flour with trace-rutinosidase flour within 65 min. These methods do not require expensive apparatuses and toxic chemicals and are therefore promising for detecting and preventing contamination by rutinosidase, e.g., in food processing factories.

## 1. Introduction

Tartary buckwheat is cultivated in mainly in China, Nepal, eastern Europe, and Japan. One important trait of Tartary buckwheat is the high rutin concentration in its seeds [1,2], which is approximately 100 times higher than that of common buckwheat seeds. In addition, buckwheat is the only known cereal to contain rutin in its seeds. The rutin concentration in Tartary buckwheat flour is approximately 2% (*w*/*w*). Therefore, Tartary buckwheat is generally used as a functional food and/or subsidiary material [3,4,5,6]. Foods that contain rutin have many beneficial functions, such as antioxidant [7,8,9] and antihypertensive activity [10], blood capillary strengthening [11,12], and alpha-glucosidase inhibitory activities [13]. In addition, the clinical effects of rutin were recently reported, and reductions in serum myeloperoxidase and cholesterol levels [14], headache, mucosal symptoms, and tiredness [15] were found in a double-blind crossover study.

However, Tartary buckwheat seeds also have high rutinosidase activity (Figure 1), which is sufficient to hydrolyze rutin in seeds within a few minutes after the addition of water [2,16].

Rutin is a functional compound. Therefore, the hydrolysis of rutin is a disadvantage of Tartary buckwheat. In addition, the hydrolysis of rutin triggers the generation of a strong bitter taste in subsequent food products [2,17,18,19].

Food-processing technology, such as the heat treatment of flour, can prevent rutin hydrolysis. However, strong and long-term heat treatment is needed to inactivate rutinosidase, and this treatment leads to serious deterioration of taste, texture [20], color, and flavor as well as added costs. Against this background, a new Tartary buckwheat variety, “Manten-Kirari” (MK), has been developed [18,19]. This variety has only trace-rutinosidase activity and is suitable for the production of rutin-rich foods without bitterness. Compared to traditional Tartary buckwheat varieties, such as ‘Hokkai T8’ (T8), the rutinosidase activity in MK seeds is two or three orders of magnitude lower [19]. Therefore, bread [21] and noodles [5,22] made with this variety have high rutin concentrations and reduced bitterness. However, the rutinosidase activity of MK is not zero but trace. Therefore, the trace-rutinosidase activity in dough gradually hydrolyzes rutin during dough storage. In addition, seed or flour contamination of the normal rutinosidase varieties, such as T8 to MK, causes rutin hydrolysis. Therefore, we can evaluate the presence or absence of contamination using HPLC. However, most factory floors do not have HPLC systems because they are expensive apparatuses. In addition, toxic solvents, such as methanol and/or acetonitrile, are needed to analyze rutin using HPLC. These materials are not suitable for handling in food processing factories in terms of safety. Against this background, we developed a simple detection method to evaluate the contamination rate using dough color with alum (potassium aluminum sulfate) solution. Alum is the generic name of the double salt of sulfate of the univalent positive ion and sulfate of metal ions of three values. Alum (potassium aluminum sulfate) is dissociated into potassium ions, aluminum ions, and sulfate ions in water, and it is used in various applications, such as food additives, pharmaceutical raw materials, and dyeing ad agents. As food additives, alum is used as a raising agent and a color retention agent. The yellow color of the flavonoid pigment is maintained by the binding of flavonoid pigments and aluminum in the alum. Therefore, we hypothesized that alum-flavonoid complexation should be a good indicator of rutinosidase activity by comparing the color of them without expensive apparatus. JECFA established a provisional tolerable weekly intake (PTWI) of 2 mg/kg bw (body weight) for aluminum [23]. (1) A survey by the Ministry of Health, Labor, and Welfare (Japan) confirmed that the estimated intake of adults is below the PTWI; and (2) alum is a relatively safe substance. Therefore, a simple detection method using the alum solution is safe and effective. In addition, the normal rutinosidase variety sometimes contaminates the trace-rutinosidase variety in the seeding step and harvesting step. Therefore, in the field where trace-rutinosidase Tartary buckwheat is cultivated, it is important to eliminate contamination from normal rutinosidase individuals. To eliminate the normal rutinosidase individuals, farmers can identify the individual in the early growth stage of ripening.

These methods do not use HPLC but rather employ common family cups, reasonable plastic tubes, or microtiter plates with nontoxic solutions. In this paper, we described the detailed procedure and effectiveness of the methods.

## 2. Results and Discussion

### Rapid Detection of Rutinosidase Activity during Ripening

In the control, which contained only the rutinosidase reaction mixture, precipitation did not occur (Figure 2), which indicates that rutin itself does not precipitate under this condition. In ripening seeds of MK, precipitation did not occur at any DAP. Apparent precipitation occurred at every DAP of T8. HPLC analysis of the precipitants revealed that most of the precipitants were quercetin (Figure 3A). For the UV detection, the rutinosidase reaction solution in T8 had yellow spots, whereas MK and the control had black spots. In ripening seeds subjected to simple detection (microplate detection), we can also distinguish MK and T8 as same as perennial buckwheat in every DAP sample (Figure 2). Not only flour milling companies, but also farmers need to evaluate contamination of the normal rutinosidase variety with MK before shipment because a risk of contamination occurs during the sowing, harvesting and drying processes. When farmers suspect that a ripening stage of the Tartary buckwheat plant in the field is trace rutinosidase or the normal rutinosidase plant, they can identify whether the plant is MK or a normal rutinosidase variety only within a few minutes by this method in the field and can eliminate the plant if it is a normal rutinosidase activity type.

First, we confirmed that steam-heat treatment to flour completely inactivates rutinosidase using HPLC. After 16 h of storage of intact MK dough, rutin was completely hydrolyzed (Figure 3B), whereas steam-heated MK did not undergo rutin hydrolysis (Figure 3C). Therefore, heat-steamed MK flour is usable as a standard of non-rutinosidase flour.

The effects of alum against rutin hydrolysis are shown in Figure 4. For intact MK dough stored for 16 h mixed with alum solution, both the dough and supernatant turned yellow (Figure 4A), whereas the heat-steamed MK did not have a yellow dough color (Figure 4B). In addition, we observed yellow precipitants on the bottom of the supernatant (Figure 4A, C arrow), which were precipitated quercetin (Figure 3D). However, these precipitants are not noticeable and therefore not suitable for an index of rutinosidase activity indicator. In intact dough stored for 16 h and heat-steamed MK mixed with non-alum solution, neither dough nor supernatant had a yellow color. (Figure 4B,C). These data indicate that the alum solution is useful for detecting rutinosidase activity by color. In addition, aluminum complexation with rutin has been reported, and the molecular modeling of an aluminum-rutin complex has been proposed [24]. Based on this modeling, we showed a possible aluminum complexation of rutin and quercetin with alum (Figure 1B,C). The above data indicate that this complexation did not prevent rutinosidase activity.

The results of the application of the method to evaluate contamination of MK flour by normal rutinosidase flour are shown in Figure 5A. Fifteen minutes after the addition of the alum solution, the dough turned yellow in the experimental plot over sevenfold of relative rutinosidase activity (Figure 5A). In addition, the supernatant became muddy in these experimental plots (Figure 5A). Sixty-five minutes after the addition of the alum solution, it was possible to distinguish the experimental plots between one- and three-fold relative rutinosidase activity by observing dough with muddy supernatants (Figure 5A). In addition, to evaluate differences of color in supernatant objectively, we also measured l*, a*, and b* in the supernatant at the contamination rate of 0–1, 65 min from the picture in open black box in Figure 5A. The contamination rate of 0–1 at 65 min is suitable for the judgement of contamination by muddiness observation. We confirmed that the l*, a*, and b* were significantly different in each experimental plot (Figure 5B). This result reinforces our hypothesis that alum-flavonoid complexation should be a good indicator of rutinosidase activity by comparing the color. In addition, this method does not use toxic substances and expensive apparatus. Therefore, it is suitable for practical application, such as in a food processing factory and/or flour milling company. The three-fold relative rutinosidase activity corresponded to 1% contamination of normal rutinosidase flour with MK flour. This 1% contamination hydrolyzed 64.8% of rutin in noodle 1 h after the addition of water [25]. Recently, Suzuki et al. [22] demonstrated that usage of NaHCO_3_ with low-temperature dough storage can prevent rutin hydrolysis with up to 1% contamination of MK flour by T8 flour. This method, with the simple and rapid detection described in this paper, produces rutin-rich foods with reduced bitterness efficiently.

## 3. Materials and Methods

### 3.1. Plant Materials

Two Tartary buckwheat varieties, trace-rutinosidase variety MK and normal rutinosidase variety T8, were sown in early June in the experimental field of Memuro Hokkaido Japan (42°52′54′′N, 143°03′18′′E, altitude 131 m). In late August, we harvested mature plants and dried them at 35 °C for approximately one week. Harvested plants were threshed, and seeds were purified using a grain fan, then subjected to stone removal. After that, seeds were stored at 10 °C.

To evaluate rutinosidase activity during ripening in Tartary buckwheat, we observed the seed set every day at maturation and marked flower buds when we found seed set, which occurred at 3, 6, 12, 18, and 24 days after pollination (DAP). We used T8, MK, and perennial buckwheat seeds for the experiment. Harvested seeds were immediately applied for the simple detection of rutinosidase activity.

### 3.2. Preparation of Flour

Tartary buckwheat seeds harvested at maturation were milled using a test role mill (Quadrumat^®^ Junior, Brabender^®^ GmbH & Co., Duisburg, Germany). Flours were passed through a 2-mm mesh, and the flour milling percentage was adjusted to 63%. We have confirmed that flour milling percentage, rutin concentration, and flour milling between T8 and MK were almost the same [19].

### 3.3. Rapid Detection of Rutinosidase Activity during Ripening

Harvested seeds during ripening were cut in half from the top of the seeds in the horizon direction. The cut seeds were suspended in rutinosidase detection solution that contained 80 mg rutin/100 mL solution (60% (*v*/*v*) of 50 mM acetic acid pH 5.5: 40% (*v*/*v*) of 2-methoxy-ethanol). We used plastic tubes or 96-well microplates for evaluation. For one seed, 150 μL/well rutinosidase detection solution was added. When the seeds have normal rutinosidase activity, such as T8 seeds, generated quercetin is precipitated in the well because the solubility of quercetin is much lower than that of rutin in the solution. In the seeds with trace-rutinosidase activity such as MK, rutin hydrolysis was very slow compared to that of T8. Therefore, quercetin was not precipitated. Fifteen minutes after the addition of rutinosidase detection solution, we observed the precipitation of quercetin in the well. Compared to the T8 and MK seeds, we can clearly identify whether the tested seeds have normal rutinosidase or trace rutinosidase. For more rapid detection of rutinosidase, 0.5 μL of reaction solution was spotted on paper 5 min after the addition of rutinosidase reaction solution. The paper was irradiated with UV light from a commercially available black light (UV light; 254 nm; e.g., Handheld UV Lamp, 6 W, UVG-54, 254 nm, 115 V, Funakoshi Co., Ltd., Tokyo, Japan), and the spot color was observed. Quercetin had luminescence upon irradiation with UV light and a yellow color was observed at the spot, whereas rutin did not have luminescence (black spot). To identify the substance of the precipitate in the T8 well, the rutinosidase reaction mixture containing the precipitant was centrifuged and washed with water 3 times. Then, the precipitate was dissolved in methanol containing 20% (*v*/*v*) 0.1% (*v*/*v*) phosphoric acid. After filtration, the solution was analyzed using HPLC [26]. HPLC was performed on a 150 mm × 2 mm i.d. Cadenza CD-C18 column (Imtakt Corporation, Kyoto, Japan) at a flow rate of 0.3 mL/min. Rutin and quercetin were separated using a 0–5 min linear gradient of 0–100% solvent A (CH_3_CN/H_2_O/Trifluoroacetic Acid, 7.5:92.5:0.1) to solvent B (CH_3_CN/H_2_O/Trifluoroacetic Acid, 55:45:0.1).

### 3.4. Simple Detection of Rutinosidase Activity in Artificially Contaminated T8 Flour and MK Flour

To evaluate rutinosidase activity, we employed the alum solution to boost the yellow color of Tartary buckwheat dough. The alum forms a complex with flavonoids, such as rutin and quercetin. The alum–quercetin complex has a more concentrated yellow color than the alum-rutin complex. Therefore, we can evaluate rutinosidase activity in dough by assessing the intensity of the yellow color in dough or the supernatant of flour-alum mixed solution and comparing it with that of the standard flour, whose rutinosidase activity (contamination rate) is known. To evaluate the detection ability of rutinosidase by this method, we prepared standard flours as follows: ‘zero-rutinosidase flour’ and ‘artificial contaminated flours’, in which a known amount of T8 flour was mixed with MK flour. The relative rutinosidase activities of the flours were 0 (zero-rutinosidase flour) and 1, 3, 7, 19, 55, and 163 (artificially contaminated flours). To prepare ‘zero rutinosidase’ flour, 10 g of MK flour packed in a nonwoven bag with 1 cm thickness was steamed using a steamer for 15 min. After steaming, partly caked flour was crushed into powder using a motor. To evaluate the rutinosidase activity in the flour, 100 mL of the alum solution containing 0.1% (*w*/*v*) alum was poured into 10 g of subject flour. After mixing well, the dough was allowed to stand at room temperature (approximately 25 °C). We observed and compared the color of the dough and supernatant at 0, 1, 15, and 65 min after the addition of the alum solution. To evaluate differences of color in the supernatant, l*, a*, and b* at a contamination rate of 0–1 at 65 min were measured from the picture in open black box in Figure 4A using GIMP software (https://www.gimp.org/downloads/ accessed on 11 January 2022). The measurement was performed 10 times and mean ± SD was expressed. Means were statistically compared by Bonferroni’s multiple comparison test [27]. We employed transparent cups, namely commercially available glass cups, to easily check the color of the dough and supernatant. To investigate whether heat treatment completely inactivated rutinosidase, the rutin and quercetin concentrations in dough stored 16 h after the addition of water were investigated using HPLC as described above.

## 4. Conclusions

The developed methods can distinguish normal rutinosidase seeds from trace-rutinosidase mature seeds and ripening young seeds using a rutin solution by comparing the muddiness, which corresponds to quercetin generation. These are reasonable and effective for detecting the contamination of rutinosidase compared to PCR-based variety discrimination methods.

Detection methods based on differences in the solubility of substances, such as rutin and quercetin, are effective as simple detection methods based on visual muddiness. In addition, these methods are promising for detecting and preventing contamination by rutinosidase, e.g., in food processing factories, because they do not require expensive apparatuses and toxic chemicals.

## Figures and Tables

**Figure 1 plants-11-00320-f001:**
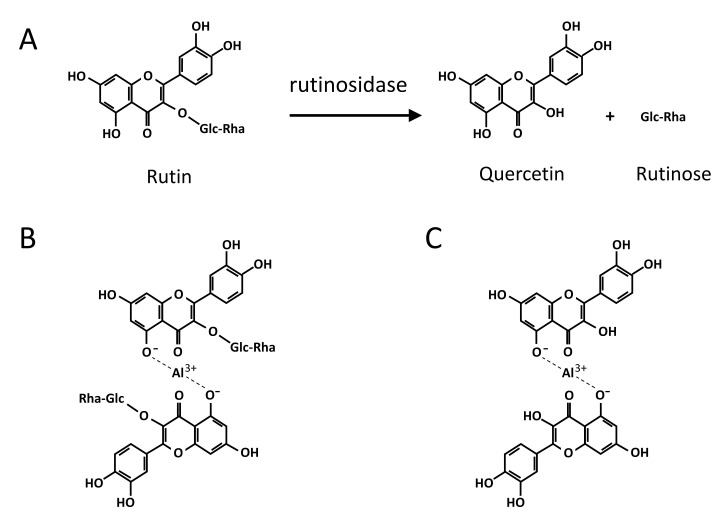
Rutin hydrolysis in Tartary buckwheat seeds and possible aluminum complexation of rutin and quercetin. (**A**) rutin hydrolysis, (**B**) aluminum complexation with rutin, (**C**) possible aluminum complexation with quercetin. (**B**,**C**) were referred from Junmei et al. 2005.

**Figure 2 plants-11-00320-f002:**
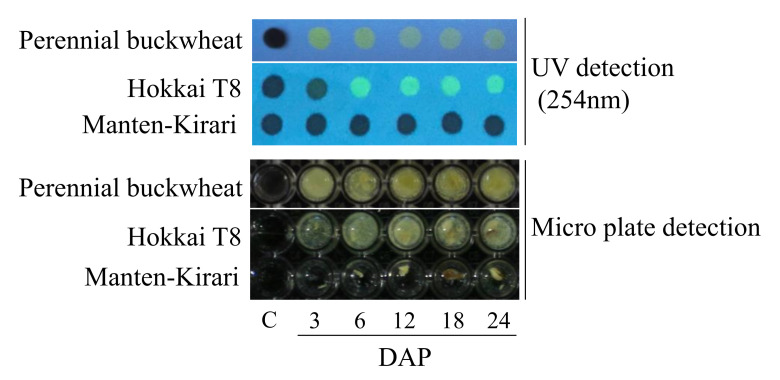
Simple detection method for rutinosidase activity in ripening and mature seeds. Harvested seeds during ripening were cut in half from the top of seeds in the horizon direction. The cut seeds were suspended in rutinosidase detection solution. For one seed, 150 μL/well rutinosidase detection solution was added. After 15 min, we observed the generation of precipitates (microplate detection). For more rapid detection of rutinosidase, 0.5 μL of reaction solution was spotted on paper 5 min after the addition of rutinosidase reaction solution. The paper was irradiated with UV light from commercially available black light (UV light; 254 nm), and the spot color (UV detection) was observed. Quercetin was luminescent upon irradiation with UV light, and a yellow color was observed at the spot. Rutin did not have luminescence (black spot).

**Figure 3 plants-11-00320-f003:**
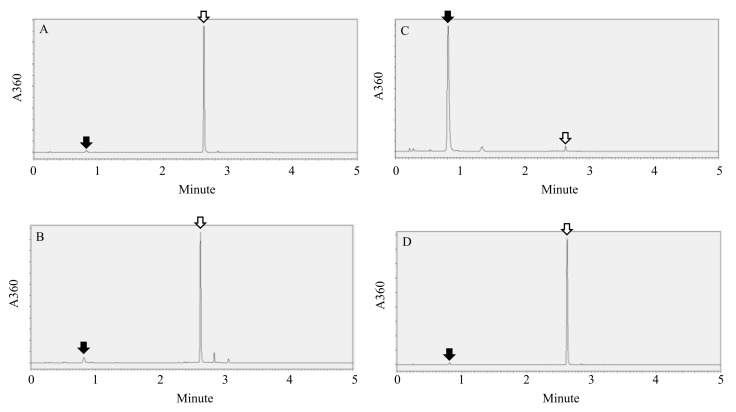
HPLC chromatogram of extract from flour and precipitant. Closed arrows indicate rutin and open arrows indicate quercetin. To identify the substance of the precipitate in the T8 well, the rutinosidase reaction mixture containing the precipitant was centrifuged and washed with water 3 times. Then, the precipitate was dissolved in methanol containing 20% (*v*/*v*) 0.1% (*v*/*v*) phosphoric acid. After filtration, the solution was analyzed using HPLC (**A**). To confirm that the steam steam-heat treatment to flour completely inactivates rutinosidase, dough extract of intact flour (**B**) and steam-heated flour (**C**) was analyzed using HPLC. The extract of yellow precipitants on the bottom of the supernatant (Figure 4A,C arrow) was also analyzed using HPLC (**D**).

**Figure 4 plants-11-00320-f004:**
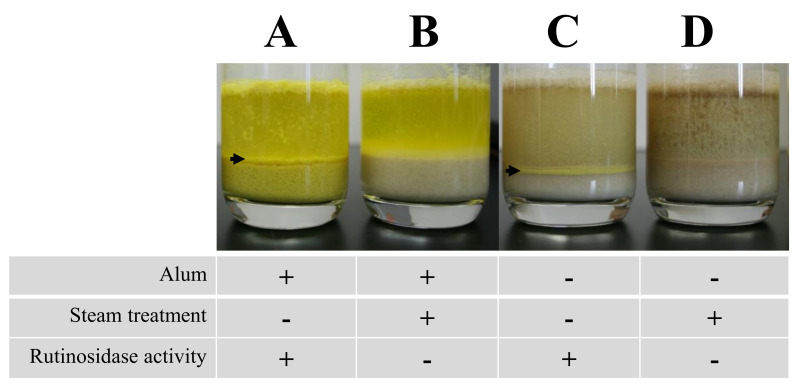
Comparison of color between water and alum solution at 16 h after their addition. To evaluate rutinosidase activity in flour, 100 mL of the alum solution containing 0.1% (*w*/*v*) alum was poured into 10 g of the investigated flour. After mixing well, the dough was allowed to stand at room temperature (approximately 25 °C). To prepare ‘zero rutinosidase’ flour, MK flour packed in a nonwoven bag was steamed using a steamer for 15 min. To investigate whether heat treatment completely inactivated rutinosidase, the rutin and quercetin concentrations in dough stored 16 h after the addition of water were investigated using HPLC.

**Figure 5 plants-11-00320-f005:**
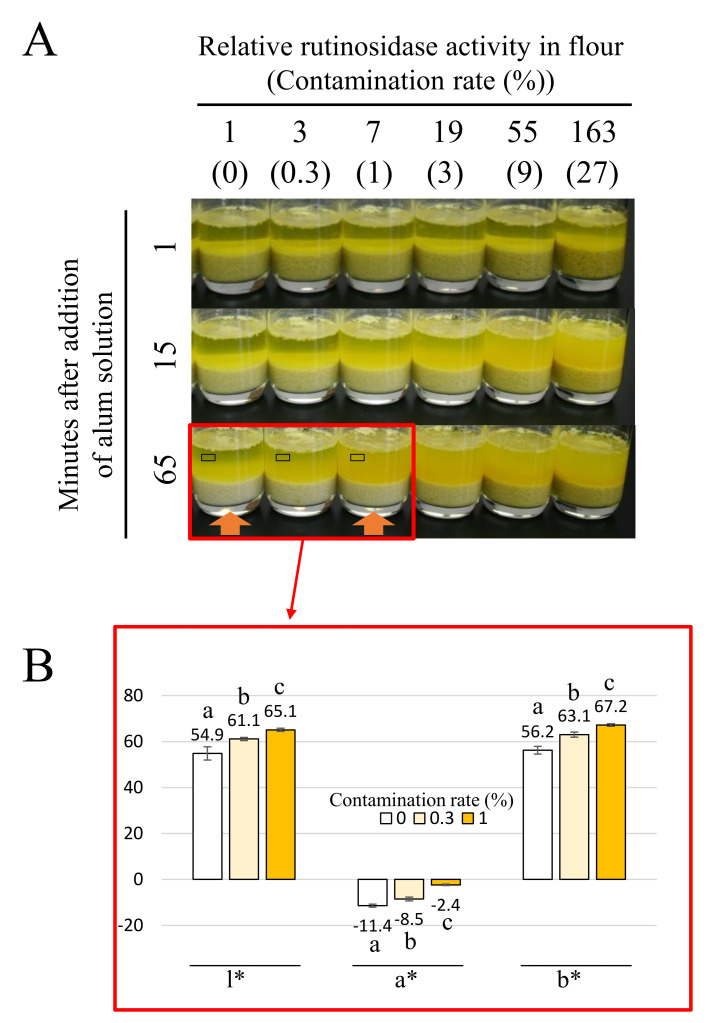
Changes in dough and supernatant color in the alum solution at different rutinosidase activities. (**A**) To evaluate rutinosidase activity in flour, 100 mL of the alum solution containing 0.1% (*w*/*v*) alum was poured into 10 g of the investigated flour. After mixing well, the dough was allowed to stand at room temperature (approximately 25 °C). We observed and compared the color of the dough and supernatant at 0, 1, 15 and 65 min after the addition of the alum solution. To evaluate the detection ability of rutinosidase by this method, we prepared standard flours as follows: ‘zero-rutinosidase flour’ and ‘artificial contaminated flours’, in which a known rate of T8 flour was mixed with MK flour. The relative rutinosidase activities of the flours were 0 (zero-rutinosidase flour) and 1, 3, 7, 19, 55 and 163 (artificially contaminated flours). (**B**) To evaluate differences of color in supernatant, l*, a* and b* in contamination rate of 0 to 1 at 65 min were measured from the picture in open black box in Figure 5A using GIMP soft wear. The measurement was performed 10 times and means ± SD was expressed. Different alphabetical character (a, b, c) indicates significant difference in average at 0.1% level in by Bonferroni’s multiple comparison test.

## Data Availability

The data that support the findings of this study are available on request from the corresponding author. The data are not publicly available due to privacy or ethical restrictions.

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
