# Peer review of "Development of Novel Detection Method for Rutinosidase in Tartary Buckwheat (Fagopyrum tataricum Gaertn.)"

_plants, 2022, doi:10.3390/plants11030320_

Round 1

Reviewer 1 Report

  1. Why did you submit a revised manuscript? Do you want to delete Figure 1 or not?
  2. The method proposed by the authors is really convenient, but the study is not very systematic.
  3. The authors did not provide the results of the quantitative analysis or the statistical results.
  4. I think this is more of a lab report than a research article.

Author Response

Response to Reviewer 1 Comments

Thank you very much again for the thoughtful feedback you provided regarding our manuscript. We also appreciate the time and effort you have dedicated to providing insightful feedback on ways to strengthen our paper. Thus, it is with great pleasure that we resubmit our article for further consideration. Please refer detail in attached word file.

Reviewer 2 Report

This paper presents a simple and rapid detection method of rutinosidase during ripening seed and dough in Tartary buckwheat (Fagopyrum Tataricum Gaertn.) in a form of a Communication.

No additional comments.

Author Response

Response to Reviewer 2 Comments

Point 1:

This paper presents a simple and rapid detection method of rutinosidase during ripening seed and dough in Tartary buckwheat (Fagopyrum Tataricum Gaertn.) in a form of a Communication.

No additional comments.

Response 1:

Thank you very much again for the thoughtful and constructive feedback you provided regarding our manuscript. We also appreciate the time and effort you dedicated to providing insightful feedback on ways to strengthen our paper.

Reviewer 3 Report

The main aim of the paper prepared by Suzuki et al. is very ambitious and can be used in practice. Authors graphically described the results and confirmation of methods useful. However, in good practice science, we also need to have some numbers results and statistical analysis. This is the main issue of the current paper, and the Authors need to correct it. Other minor issues:

In the abstract, please add more specific information about the developed method.

Please correct all track changes

In figure 1 legend, please add more information about the demonstrated process

I suggest putting figure 1 at the end of the introduction section.

According to the results presented in figure 2, please also add a table with the number value of absorbance.

Please put HPLC results which you obtained

Author Response

Response to Reviewer 3 Comments

Thank you very much again for the thoughtful feedback you provided regarding our manuscript. We also appreciate the time and effort you have dedicated to providing insightful feedback on ways to strengthen our paper. Thus, it is with great pleasure that we resubmit our article for further consideration. Please refer detail in attached word file.

Reviewer 4 Report

The research is interesting in the area of the rutinosidase determination during ripening seed and dough in Tartary buckwheat (Fagopyrum Tataricum Gaertn.). Nevertheless, the manuscript needs to be improved in order to provide enough information to justify its importance and novelty.

Please revise the title to be more attractive and clear to read.

Please re-structure the section background in order to explain the novelty of the research, the research question, hypothesis and the objectives.

Please include informations and HPLC chromatograms on rutinoside determination.

Please conclude according to the main objectives of the research and include more practical application on your findings.

Author Response

Response to Reviewer 4 Comments

Thank you very much again for the thoughtful feedback you provided regarding our manuscript. We also appreciate the time and effort you have dedicated to providing insightful feedback on ways to strengthen our paper. Thus, it is with great pleasure that we resubmit our article for further consideration. Please refer detail in attached word file.

Round 2

Reviewer 1 Report

The paper has been improved. 

Reviewer 3 Report

All my comments have been included in the new version of manuscript. I accept the paper.

Reviewer 4 Report

The manuscript has been improved.